# Adolescents’ Bipolar Experiences and Suicide Risk: Well-being and Mental Health Difficulties as Mediators

**DOI:** 10.3390/ijerph18063024

**Published:** 2021-03-15

**Authors:** Ascensión Fumero, Rosario J. Marrero, Alicia Pérez-Albéniz, Eduardo Fonseca-Pedrero

**Affiliations:** 1Department of Clinical Psychology, Psychobiology and Methodology, University of La Laguna, 38200 La Laguna, Tenerife, Spain; rmarrero@ull.edu.es; 2Department of Educational Sciences, University of La Rioja, 26002 Logroño, La Rioja, Spain; alicia.perez@unirioja.es (A.P.-A.); eduardo.fonseca@unirioja.es (E.F.-P.)

**Keywords:** adolescents, suicide risk, bipolar disorder, emotional problems, peer problems, well-being

## Abstract

Bipolar disorder is usually accompanied by a high suicide risk. The main aim was to identify the risk and protective factors involved in suicide risk in adolescents with bipolar experiences. Of a total of 1506 adolescents, 467 (31%) were included in the group reporting bipolar experiences or symptoms, 214 males (45.8%) and 253 (54.2%) females. The mean age was 16.22 (*SD* = 1.36), with the age range between 14 and 19. Suicide risk, behavioral and emotional difficulties, prosocial capacities, well-being, and bipolar experiences were assessed through self-report. Mediation analyses, taking gender as a moderator and controlling age as a covariate, were applied to estimate suicide risk. The results indicated that the effect of bipolar experiences on suicide risk is mediated by behavioral and emotional difficulties rather than by prosocial behavior and subjective well-being. Specifically, emotional problems, problems with peers, behavior problems, and difficulties associated with hyperactivity were the most important variables. This relationship was not modulated by gender. However, the indirect effects of some mediators varied according to gender. These results support the development of suicide risk prevention strategies focused on reducing emotional difficulties, behavioral problems, and difficulties in relationships with others.

## 1. Introduction

Adolescence involves a set of physical, cognitive, and social changes as well as mood fluctuations that can increase vulnerability toward psychological tension, crises, or mental health problems [1,2]. Suicidal behavior is currently one of the leading causes of death among adolescents [3]. Of European adolescents, 4.2% report suicidal ideation and suicide attempts [4]. The empirical results show that suicidal behavior is related to multiple risk factors [5,6] and psychopathological problems [7,8]. Suicide attempts are associated with substance or alcohol abuse, depressive polarity of the first or most recent episode, and comorbidities such as cluster B personality disorders [9]. Moreover, suicide risk is more often found in female adolescents and early adolescence [10].

During adolescence, the individualization process involves young people negotiating their independence from their parents and establishing equal relationships with their peers [11], as well as beginning to spend more time with their peers [12]. Adolescents spend a lot of time in school and establish a significant proportion of their interpersonal relationships in this setting [13]. Peers serve as models for the management of different situations and can be a source of interpersonal difficulties, compromising young people’s emotional adjustment and well-being [14,15,16]. Negative peer interactions have been associated with suicidal ideation in youth [17]. In addition, it has been found that peer and emotional problems are associated with decreased quality of life, which in turn raises suicide risk [18].

A great deal of research has established that the most common symptoms reported by adolescents include emotional and behavioral problems. The prevalence of emotional and behavioral problems among adolescents ranges from 17% to 30% in Asian countries [19,20], and is around 45.7% in Europe [21]. As well, anxiety disorders (31.9%), followed by behavior disorders (19.1%), are the most prevalent disorders in U.S. adolescents [22]. In Spain, approximately 4% of a sample of children and adolescents evaluated through the Spanish National Health Survey presented some type of emotional or behavioral problems [23]. Emotional dysregulation at the affective (anxiety and depression), cognitive (attentional problems), and behavioral levels (aggressiveness) has been linked to suicide risk [24,25]. Some studies have shown that emotional difficulties have both direct and indirect effects on suicide risk [26,27,28]. Behavioral problems have been associated with a high probability of suicide risk. In this sense, children’s suicide risk behavior is predicted fundamentally by changes produced by traumatic experiences in the early personality, such as aggressiveness and impulsivity [29,30]. In adolescents, both behavioral and emotional difficulties are associated with increased suicide risk [31,32,33,34]. Strengths, specifically prosocial behavior, protect against suicide risk while difficulties, such as emotional or behavioral problems, are among the best predictors of self-harm and suicidal ideation [35,36].

Additionally, hyperactivity and internalizing and externalizing symptoms occur more frequently in suicidal groups than in controls [37]. The effects of hyperactivity on suicidality, in general, remain after considering the mediating roles of family function and psychiatric comorbid conditions. For instance, the excess risks of suicidality (i.e., suicidal ideation, suicide planning, and suicide attempts) showed that the mediating effects of emotional and behavioral problems are higher across all suicide outcomes than the mediating effects of family function in hyperactive children [38].

Furthermore, it has become particularly important to analyze mental health not only from a risk-based approach but also from an approach focused on protective factor as well-being, and quality of life [39]. Subjective well-being (SWB) appears to be positively associated with physical health [38,39,40,41], personal characteristics [42,43], adolescents’ perceived parental involvement [44], and greater peer support and feeling safe at school [45]. Negative associations have been found between SWB and behavioral problems [46], psychological disorders [47], and suicidal behavior [48]. Individuals with high levels of pathological functioning exhibit lower levels of subjective well-being [49].

Moreover, prosocial behavior has been associated with well-being [50]. Prosociality emerges early in the human being from the bonds established between caregivers and infants and focuses on helping and sharing [51]. Helping others establishes social connections, increases social acceptance, and is associated with higher levels of positive emotions and meaning in life [52,53]. Previous research has found that peer groups can influence prosocial behavior in adolescence, generating links with others that promote better social adjustment [54] and decrease the likelihood of behavior problems [55] and, possibly, also suicide risk.

Suicide rates are especially high in psychological disorders characterized by instability or emotional dysregulation, such as bipolar disorder or borderline personality disorder [56]. About one quarter of children and young people with bipolar disorder are at high risk of attempting suicide and between 50% and 57% show suicidal ideation [57]. In a five-year longitudinal study, 18% of youths with bipolar disorder reported attempting suicide [58]. Young people with bipolar disorder who have attempted suicide show lower quality of life than those who have only manifested suicidal ideation [57]. The importance of suicide attempts in persons with bipolar disorder is based on the fact that it is a solid predictor of death by suicide [59,60] and the main cause of premature mortality in patients with bipolar disorder [61]. Bipolar spectrum disorders are difficult to identify in youth due to the temperamental instability and mood changes associated with this life stage. It is difficult to choose between including all forms of affective instability under the same general class of bipolar spectrum disorder or splitting them into categories [62]. Subthreshold bipolar disorder is common, and the variation in the number of times a symptom is experienced suggests a continuum of bipolar severity, from cyclothymia to bipolar II and bipolar I disorder [63]. In our study, bipolar experiences or symptoms are defined as the occurrence of seven or more items of the 13 proposed in a self-report measure of manic episodes. Only manic and hypomanic symptoms have been considered, since the aim of this study was to identify the symptomatology at the subclinical level. Although a major depressive episode may appear in bipolar spectrum disorders, in the case of bipolar I disorder it is sufficient that the manic episode be present. For this reason, only manic symptoms have been included, and depressive symptoms have been excluded. Previous research has shown that the Mood Disorder Questionnaire (MDQ) could be used as a screening tool to evaluate hypomanic experiences in the non-clinical adolescent [64]. A positive MDQ screen is associated with greater perceived stress, more family, social and occupational difficulties, and poor quality of life [65]. Subclinical manic experiences, considered jointly with other risk markers, could help to predict bipolar spectrum disorders later. Bipolar experiences, as psychotic-like experiences [66], would be located as a point on the continuum next to the transitory intermediate states or subclinical processes where the number of times a symptom is experienced may increase the liability to suffer a bipolar disorder. Through self-reports, adolescents may provide information on symptoms for a manic or hypomanic episode, but diagnosis cannot be made based solely on the mood state [67].

Although a large volume of evidence has been accumulated on the determinants of bipolar disorder, suicide risk, personal difficulties, and well-being in adults, there are hardly any studies that analyze this issue in adolescents [36]. In this study, multiple protective and risk factors for suicidal behaviors were analyzed simultaneously. We propose to test if the emotional, cognitive, behavioral, and social difficulties as well as protective factors related to prosocial behavior and subjective well-being will contribute to estimating suicide risk in adolescents with bipolar experiences or symptoms.

## 2. Materials and Methods

### 2.1. Participants

The sample was recruited from 34 schools and 98 classrooms in the region of La Rioja, Spain. To guarantee a representative sample, stratified random sampling was used. Participants were selected using a stratified cluster sampling design, with the classroom as the sampling unit, from a population of 15,000 students. The variables used to stratify the sample were school status (public or private) and school stage (secondary, baccalaureate, or vocational training). The assignment of the classrooms was proportional to the number of students. For the estimation of the sample size, the G power program (Heinrich Heine Universität, Düsseldorf, Germany) was used (http://www.psychologie.hhu.de/arbeitsgruppen/allgemeine-psychologie-und-arbeitspsychologie/gpower.html; accessed on 10 July 2017). An error of 5% was considered, with a sample size of 15,000 and a confidence level of 85%. Therefore, a total of 1478 participants were necessary. There were 1845 students in the initial sample, although some participants were excluded for obtaining high scores on the infrequency scale (more than three points) (*n* = 104), being older than 19 years of age (*n* = 170), or not completing all the administered self-reports (*n* = 65). The final sample was 1506 students from 14 to 19 years of age (*M* = 16.15, *SD* = 1.36), 55.7% of whom were female. With regards to nationality, the distribution was as follows: 89.9% Spanish, 3.7% Latin American (Bolivia, Argentina, Colombia, and Ecuador), 0.7% Portuguese, 2.4% Romanian, 1% Moroccan, 0.7% Pakistani, and 2% other nationalities.

### 2.2. Instruments

Strengths and Difficulties Questionnaire (SDQ) [68]. The SDQ has been used as a tool for the screening and epidemiological analysis of children’s and adolescents’ mental health [69,70]. The SDQ consists of 25 statements with a Likert-type response format containing three options (0 = no, the individual has not experienced it; 1 = sometimes the individual has experienced it; and 2 = yes, the individual has experienced it) about how the person has felt in the last six months. The items are distributed across five subscales: Emotional symptoms, behavioral problems, hyperactivity, peer problems, and prosocial behavior. The first four subscales yield a Total Difficulties score. The higher the score, the greater the level of emotional and behavioral difficulties, except for the subscale of prosocial behavior, where a lower score corresponds to worse adjustment. In the present study, the Spanish version validated in adolescents was used [23]. Cronbach’s alpha for the Total Difficulties score was 0.75.

Personal Well-being Index–School Children (PWI–SC) [71]. The PWI–SC items assess satisfaction with seven domains: Standard of living, health, life achievements, relationships, safety, community-connectedness, and future security measured on a ranging from “completely dissatisfied” (0) to “completely satisfied” (10). The score on the global scale is the result of adding up the scores on these seven items. Therefore, the total score can range from 0 to 70 points. The PWI–SC has shown adequate psychometric properties in previous international and national studies [71,72]. The Spanish version validated in adolescents was used [73]. In the current study, Cronbach’s alpha for the total score was 0.83.

Paykel Suicide Scale (PSS) [74]. This 5-item (yes/no) questionnaire assesses suicide risk during the last year. Two items address suicidal ideation, two others ask about serious suicidal plans, and one asks about suicide attempts. The score on the global scale is the result of adding up the scores on these five items. Higher scores are related to greater risk and severity of suicide. In this study, the overall score on the Paykel scale was used. The psychometric properties of the Spanish PSS have been examined in previous studies [36]. Cronbach’s alpha was 0.93.

Mood Disorder Questionnaire [75]. This consists of 13 yes/no items based on the Diagnostic and Statistical Manual of Mental Disorders, Fifth Edition (DSM–5) criteria for bipolar disorder [76]. The instrument refers exclusively to manic symptomatology, manic polarity, or hyperthymic symptomatology. A result is considered positive if the participant replies affirmatively to seven or more items of the 13 proposed (criterion 1) at the time of assessment. In addition, the symptoms described must have occurred during the same time period (criterion 2) and represented moderate or severe problems (criterion 3). In this study, to classify the adolescents with bipolar experiences or symptoms, only criterion 1 was considered. The Spanish version validated in adolescents and young adults was used [64], with a reliability of 0.85.

Oviedo Infrequency Scale (INF–OV) [77]. This was administered to participants to detect those who responded in a random, pseudorandom, or dishonest manner. The INF–OV instrument is a self-report composed of 12 items in a 5-point Likert scale format (1 = completely disagree; 5 = completely agree). Students with more than three incorrect responses were eliminated from the sample.

### 2.3. Procedure

The research was approved by the Educational Government of La Rioja and the Ethical Committee of Clinical Research of La Rioja (CEICLAR). The tests were administered collectively, through personal computers, in groups of 10 to 30 students, during normal school hours and in a classroom specially prepared for this purpose, under the supervision of researchers trained in a standard protocol. No incentive was provided for participation. For participants under 18, parents provided written informed consent for their child to participate in the study. Participants were informed of the confidentiality of their responses and of the voluntary nature of the study.

### 2.4. Data Analysis

A cross-sectional study design was used. First, chi-square was used to check for gender and age differences between the two groups, that is, participants with (score equal to or greater than 7) and without bipolar experiences. Second, the mean differences between the two groups for the variables included in the study were analyzed. Third, we analyzed the relationships between sociodemographic variables and suicide risk with all the variables included in the study (SDQ subscales and well-being) in adolescents with bipolar experiences, using Pearson’s correlation for continuous variables and Spearman’s correlation for ordinal variables. Fourth, the interaction of the centered age variable with each of the model predictors (SDQ subscales and well-being) and suicide risk through multiple regressions was analyzed. Thus, it was decided whether to include age in mediation models as a moderating or covariant variable.

Finally, different mediation analyses were used to test the hypothesis that risk and protective factors mediated the relationship between bipolar experiences and suicide risk. Mediation analysis can estimate indirect and direct effects and the proportion mediated, a statistical measure estimating how much of the suicide risk works through mediators. The represented directionality was based on strong cross-sectional correlations between symptoms [78]. Gender was included as a moderator of the relationship between difficulties, well-being, and suicide risk and age was included as a covariate. In each analysis, the specific indirect effects from 1000 bootstrap samples were estimated. The statistical model appears in Figure 1, which includes gender as a moderator and the interaction with both mediator variables. First, a model was tested in which well-being and total difficulties (SDQ) were considered as mediators. Next, five other models were tested in which well-being was maintained and each subscale of the SDQ was included to determine which had a greater effect on the suicide risk. The means of the variables were centered to estimate the parameters. Gender was coded 0 for boys and 1 for girls. The analyses used the boys as a reference, although indirect effects were obtained separately for boys and girls.

The *R* program [79,80] was used, through ULLRToolbox [81].

## 3. Results

Of the total participants, 69% (*N* = 1039) had no bipolar experiences or symptoms. The mean age was 16.11 years (*SD* = 1.37) and 56.4% were female. The remaining participants (31%, *N* = 467) had bipolar experiences or symptoms. In this group with bipolar experiences, the mean age was 16.22 (*SD* = 1.36) and 54.2% were female. Of the participants with bipolar experiences, 50.7% answered affirmatively at least one item of the suicide risk scale. Specifically, 46.9% reported suicidal ideation, 26.6% had planned suicide and 6% reported suicide attempts, or in other words, answered at least one positive item in each subscale.

### 3.1. Preliminary Analysis

The authors can provide these data on request. Distributions of variables were evaluated for normality through Kolmogorov–Smirnov (K–S) and were found to be non-normal for all variables for both groups without and with bipolar experiences. For adolescents without bipolar experiences or symptoms, the obtained results were: Suicide risk K–S = 0.364, *p* < 0.001; emotional symptoms K–S = 0.134, *p* < 0.001; behavioral problems K–S = 0.187, *p* < 0.001; peer problems K–S = 0.233, *p* < 0.001; hyperactivity K–S = 0.113, *p* < 0.001; prosocial behavior K–S = 0.234, *p* < 0.001; subjective well-being K–S = 0.107, *p* < 0.001; and bipolar experiences K–S = 0.156, *p* < 0.001. For adolescents with bipolar experiences or symptoms, the obtained results were: Suicide risk K–S = 0.279, *p* < 0.001; emotional symptoms K–S = 0.121, *p* < 0.001; behavioral problems K–S = 0.168, *p* < 0.001; peer problems K–S = 0.219, *p* < 0.001; hyperactivity K–S = 0.096, *p* < 0.001; prosocial behavior K–S = 0.204, *p* < 0.001; subjective well-being K–S = 0.105, *p* < 0.001; and bipolar experiences K–S = 0.241, *p* < 0.001. Certainly, non-normal data are frequently encountered in applied social science research [82]. However, the violation of the normality assumption has little effect on the fixed effects estimates in mediation analysis but do have some effect on standard errors for the random effects [83].

Preliminary analyses were carried out to identify the differences in the sociodemographic variables between the groups with and without bipolar experiences. No statistically significant differences by gender (χ^2^(1) = 4.21, *p* = 0.433) or age (χ^2^(5) = 8.28, *p* = 0.141) were found between the groups. A general linear model (GLM) was carried out comparing adolescents with and without bipolar experiences, including five SDQ subscales as dependent variables, total score in suicidal risk, and total score in subjective well-being (Table 1). Results showed statistically significant differences between the two groups in emotional symptoms (*F*(1, 1504) = 21.04, *p* < 0.001, η^2^ = 0.01, 1 − β = 0.99), behavioral problems (*F*(1, 1504) = 106.95, *p* < 0.001, η^2^ = 0.07, 1 − β = 1), hyperactivity (*F*(1, 1504) = 77.39, *p* < 0.001, η^2^ = 0.05, 1 − β = 1), peer problems (*F*(1, 1504) = 17.06, *p* < 0.001, η^2^ = 0.01, 1 − β = 0.98), and suicide risk (*F*(1, 1504) = 24.10, *p* < 0.05, η^2^ = 0.02, 1 − β = 0.99). Marginally significant differences were found in subjective well-being (*F*(1, 1504) = 3.41, *p* = 0.065, η^2^ = 0.002, 1 − β = 0.46). Adolescents with bipolar experiences reported higher emotional, behavioral, and peer difficulties; hyperactivity; and suicide risk, and lower subjective well-being.

Correlations were analyzed between sociodemographic variables (gender and age), suicide risk, and the remaining variables in adolescents with bipolar experiences. Results are depicted in Table 2. There was a significant positive association between female gender and total difficulties, emotional symptoms, and suicide risk and a significant negative association with subjective well-being. The youngest ages were strongly related to behavioral problems and suicide risk. Additionally, there was a significant positive association between suicide risk and all included variables of the emotional and behavioral difficulties, bipolar experiences, and subjective well-being. Bipolar experiences showed no significant correlations with age or gender.

The results of multiple regression analyses of the interaction of the centered age variable with each of the centered model predictors (SDQ subscales and well-being) and suicide risk showed that these interactions were not significant: Total difficulties X age (β = 0.007, *p* = 0.38), emotional symptoms X age (β = −0.001, *p* = 0.96), behavioral problems X age (β = 0.018, *p* = 0.73), peer problems X age (β = 0.036, *p* = 0.19), hyperactivity X age (β = −0.012, *p* = 0.59), prosocial behavior X age (β = −0.041, *p* = 0.20), and well-being X age (β = −0.005, *p* = 0.38).

### 3.2. Mediation Analyses

The proposed general mediation model examined whether total difficulties (risk factors) and prosocial behavior and subjective well-being (protective factors) mediated the relationship between bipolar experiences and suicide risk for adolescents who reported this symptomatology based on the MDQ. Gender and age showed a prior association with some of the mediating variables and the criterion variable in the adolescents with bipolar experiences, so they were included as moderator and covariate, respectively. Table 3 shows the parameters estimated for each relationship between variables of the general mediation model with well-being and total difficulties as mediators.

Results revealed that bipolar experiences had an independent significant effect on total difficulties (*a*1) and well-being (*a*2) and both mediators (*b*1 and *b*2) had an independent significant effect on suicide risk, but the direct effect of bipolar experiences on suicide risk (*c*) was not significant. Age showed a negative effect on suicide risk. Neither gender (*c*1) nor the interaction between gender and total difficulties (*c*2) nor the interaction between gender and well-being (*c*3) had an effect on suicide risk. Only significant indirect effects of bipolar experiences on suicide risk through total difficulties were found for boys (*a*1 * *b*1 = 0.09) and girls (*a*1 * (*b*1 + *c*2) = 0.10). The indirect effects of well-being were not statistically significant for boys or girls. The total effect of bipolar experiences on suicide risk was 0.18 for girls [(*a*1 * (*b*1 + *c*2)) + (*a*2 * (*b*2 + *c*3)) + *c*] and 0.17 for boys [(*a*1 * *b*1) + (*a*2 * *b*2) + *c*]. The model explained 20.6% of the variance of suicide risk.

Table 3 shows the parameters estimated for each SDQ subscale and subjective well-being as mediators of the relationship between bipolar experiences and suicide risk. Results revealed that bipolar experiences had an independent significant effect on emotional symptoms, behavioral problems, peer problems, and hyperactivity (*a*1) and well-being (*a*2). Similarly, the results showed that each SDQ subscale (*b*1) and well-being (*b*2) had an independent significant effect on suicide risk. Prosocial behavior was the only mediator that had no significant relationships with bipolar experiences or suicide risk. Age showed a negative effect on suicide risk in all mediation models, whereas gender had no effect on suicide risk.

Indirect effects were found for emotional symptoms as a mediator of the relationship between bipolar experiences and suicide for boys only (*a*1 * *b*1 = 0.04). There also appeared a direct effect of bipolar experiences on suicide risk for boys and girls (*c* = 0.11). The indirect effects of well-being were not significant for boys or girls. The model explained 23.7% of the variance of suicide risk.

Behavioral problems were a mediator between the relationship of bipolar experiences and suicide risk with an indirect effect for boys (*a*1 * *b*1 = 0.02), whereas well-being had a significant indirect effect for girls only (*a*2 * (*b*2 + *c*3) = 0.04). A direct effect of bipolar experiences on suicide risk was significant for boys and girls (*c* = 0.12). The model explained 15.5% of the variance of suicide risk.

Peer problems had a significant indirect effect for boys (*a*1 * *b*1 = 0.04), but were not significant for girls. The indirect effects of well-being were not significant for boys or girls. A direct effect was found for bipolar experiences on suicide risk for girls and boys (*c* = 0.11). The explained variance was 16.4%.

Hyperactivity had significant indirect effects on the relationship of bipolar experiences and suicide risk for boys (*a*1 * *b*1 = 0.03), whereas well-being was a significant mediator for girls (*a*2 * (*b*2 + *c*3) = 0.04). A direct effect of bipolar experiences on suicide risk was found for boys and girls (*c* = 0.12). The model explained 15.5% of the variance of suicide risk.

The model that included both protective factors, such as prosocial behavior and well-being, as mediators on the relationship between bipolar experiences and suicide risk showed that prosocial behavior had no significant indirect effects. Direct effects of bipolar experiences on suicide risk were found (*c* = 0.14). For girls only, well-being had significant indirect effects (*a*2 * (*b*2 + *c*3) = 0.05) on the relationship between bipolar experiences and suicide risk. Higher well-being scores indicated lesser risk of suicide for girls with bipolar experiences. This model explained 14.9% of the variance of suicide risk.

## 4. Discussion

The current study analyzed the factors involved in suicide risk in adolescents with bipolar experiences, considering the mediator role of risk and protective factors. A striking result of our study was the high prevalence of young people with bipolar experiences. In adolescence there is a greater sense of invulnerability to risk accompanied by greater energy, activity, sociability, and sexual interest [84]. Some of these factors are not only evaluated as manic symptoms in the MDQ, but have also been linked to extraversion. Previous research has shown that manic symptoms are associated with higher extraversion [85]. Long-term changes in extraversion have been found to decrease across developmental phases [86]. Since extraversion occurs to a greater extent in adolescence, this could explain the high percentage of young people with bipolar symptoms in our study.

Young people with bipolar experiences exhibited more risk of suicide and more emotional and behavioral difficulties, hyperactivity, and peer problems than those who did not manifest these bipolar experiences, consistent with previous studies [57,58]. Specifically, about 50% reported suicidal ideation and a quarter had planned it. In addition, they showed lower subjective well-being, although the difference was only marginally significant. Fluctuations in mood at this stage of changes can make young people more vulnerable to suicide risk [56].

In the group of young people with bipolar experiences, the factors that showed a greater association with suicide were emotional symptoms, low subjective well-being, and gender. The girls showed more suicide risk, more emotional symptoms, and lower subjective well-being. Problems with peers also showed a moderate relationship with suicide risk. These results are in line with previous findings that quality of life mediates the relationship between emotional problems and suicide risk [18].

Similarly, emotional problems appear in girls in childhood, and these internalizing symptoms are considered to be linked to the development of long-term anxiety [87]. Age was associated with a lower probability of suicide and a decrease in behavioral problems. However, other studies have found a higher prevalence of suicide in adolescents aged 15–17 than in those aged 13–14 in low- and middle-income countries [88].

The risk factors, such as emotional and peer problems, were more potent mediators than the protective factors—subjective well-being and prosocial behavior—in the relationship between bipolar experiences and suicide. These findings are consistent with other studies on peer relationships and adolescent suicide risk [89], suggesting that experiences with peers indirectly affect suicide risk through mental health problems. Although gender had no effect on suicide risk, the indirect effects of some mediators varied for boys and girls in all mediation models.

The emotional symptoms were a significant mediator, showing a significant indirect effect for boys. This indirect effect of emotional difficulties associated with suicide risk has been shown previously [26,90]. Moreover, peer problems were a significant mediator of the relationship between bipolar experiences and suicide for boys only, while well-being did not show significant indirect effects. It is possible that peer problems are especially pernicious for boys, suggesting that greater influences on social self-worth increase suicide risk. The socialization that attributes different gender roles could contribute to boys’ exhibiting more peer problems, such as aggression and antisocial conduct, to maintain dominance, whereas girls experience more cooperative and prosocial behaviors, thus avoiding peer problems [91].

An indirect effect of behavioral problems and hyperactivity on the relationship of bipolar experiences and suicide was found for boys. Much of the relationship between childhood adversity and suicide is mediated indirectly by aggression, impulsivity, and adolescent problem behaviors [30]. The effects of hyperactivity on suicidal ideation, suicide planning, and suicide attempts remained after considering the mediating role of psychiatric comorbid conditions and family function [38].

Contrary to expectations, prosocial behavior was not a significant mediator in the relationship between bipolar experiences and suicide. It may be of interest to analyze the role of prosocial behavior before or during the manic episode to identify whether it could function as a protective or risk factor. Subjective well-being had significant indirect effects for girls only. Previous research has found that engaging in behaviors that promote positive responses and well-being, such as self-compassion or gratitude, are greater in girls than boys, and that gratitude mediates the relationship between negative life events and suicide risk [92,93,94].

Our findings suggest two paths estimating suicide risk: One associated with risk factors and another associated with protective factors. The primary path had a greater impact on boys, whereas the secondary path had a greater prevalence in girls. The gender differences reflect the way in which maladjustment is expressed [95]. In females it takes the form of internalizing problems, such as sadness and withdrawal, which could be associated with greater vulnerability to depression [96], and in males it takes the form of externalizing problems, like distractibility, impulsivity, and hyperactivity, which is associated with behavioral disorders [97,98].

The results must be taken with caution, since the study has several limitations. First, the adolescents in our study reported bipolar experiences, so they could have some vulnerability to the disorder, but diagnosis based only on symptoms identified in cross-sectional assessment appears to be insufficient for the accurate early detection of emerging bipolar disorder. Previous research indicates that when adolescent samples are included, greater prevalence is found [67]. Furthermore, a narrower definition of bipolar experiences or symptoms, including criteria 2 and 3 or taking into account depressive symptoms, would reflect lower prevalence. To develop instruments with greater sensitivity could clarify this issue [99]. Second, although bipolar experiences could be involved in the vulnerability to onset of the first mood episode in adolescents, it should be taken into account that if the first episode is depressive, the diagnosis of bipolar disorder becomes difficult [67]. Third, other clinical young samples could have been included to determine whether the risk and protective factors of suicide vary according to the young people’s symptomatology. Fourth, suicide risk was assessed globally without considering the cases of young people who had previously had suicidal ideation, planning, or attempts. It will be important to determine whether our results change when each component of suicide is analyzed separately [100]. Fifth, the data were cross-sectional, limiting conclusions that can be drawn from the mediation analyses. With mediation models, it is possible to analyze, through a sample of empirical data, if the model is consistent with the theory and estimate the parameters of an apparent cause–effect relationship. However, these analyses are performed at a specific point in time, so they do not report whether young people with bipolar symptoms could eventually develop the disorder. Besides, the temporality bias could affect the proposed model in the sense that the relationship between the variables may be the inverse. Finally, other third variables that have not been analyzed could explain the relationship.

Further research is needed to examine the process involved in the predisposition to suicide risk in diverse clinical problems to identify whether there is a common underlying cause or specific factors of disorders that make some adolescents more vulnerable than others. To detect the overlapping symptoms of a variety of adolescents with affective instability as the same sort of problem, it may be more useful to cluster these symptoms into a general class of bipolar spectrum disorder. Mental health information on the internet would be a resource for understanding and handling psychological problems in adolescents [101]. Additionally, it would be important to add to the investigation the potential mechanisms contributing to gender differences in the mediator role of risk and protective factors in the relationship between bipolar experiences and suicide. Longitudinal studies could be developed to identify which of the emotional and behavioral problems, among others, would operate as early manifestations of bipolar disorder or suicide risk.

In spite of the limitations, our findings provide insight into the role of several mediators in the relationship between bipolar experiences and suicide risk in a young sample with high probability of developing bipolar disorder. Risk factors, especially peer and emotional problems, had greater relevance in suicide risk than protective factors. These results have important implications for the development of prevention programs focused on identifying emotional or peer problems and protective factors for suicide risk. Therefore, promoting opportunities to experience positive situations and emotions in young people will contribute to greater well-being, greater protection against the negative effects of stress, and better positive development and adjustment.

## 5. Conclusions

The study analyzed the factors involved in suicide risk in adolescents with bipolar experiences, considering the mediator role of risk and protective factors. The risk factors, such as emotional and peer problems, were more potent mediators than the protective factors—subjective well-being and prosocial behavior—in the relationship between bipolar experiences and suicide risk. Additionally, risk factors had a greater impact on boys, whereas the protective factors had a greater prevalence in girls.

## Figures and Tables

**Figure 1 ijerph-18-03024-f001:**
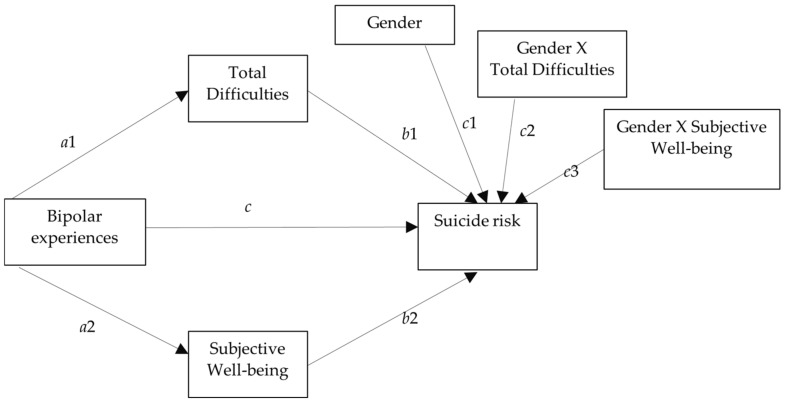
Statistical mediation model moderated by gender. The lettering describes the paths.

**Table 1 ijerph-18-03024-t001:** Mean differences between adolescents without and with bipolar experiences.

Predictors	Adolescents without Bipolar Experiences *N* = 1039		Adolescents with Bipolar Experiences *N* = 467		*F*	η^2^
	Mean	*SD*	Mean	*SD*		
Emotional symptoms	3.37	2.43	4	2.44	21.04 ***	0.014
Behavioral problems	1.68	1.48	2.59	1.82	106.95 ***	0.066
Peer problems	1.41	1.53	1.77	1.61	17.06 ***	0.011
Hyperactivity	4	2.13	5.03	2.09	77.39 ***	0.049
Prosocial behavior	8.63	1.45	8.51	1.49	2.39	-
Suicide risk	0.79	1.25	1.15	1.45	21.10 ***	0.016
Subjective well-being	55.38	9.17	54.46	8.61	3.41 *	0.002

* *p* < 0.10, *** *p* < 0.001.

**Table 2 ijerph-18-03024-t002:** Correlational analysis in adolescents with bipolar experiences (*N* = 467).

Variables	Gender ^a^	Age ^b^	Suicide Risk ^b^
Emotional symptoms	0.33 ***	−0.004	0.44 ***
Behavioral problems	−0.07	−0.13 **	0.19 ***
Peer problems	−0.03	0.03	0.29 ***
Hyperactivity	0.05	−0.05	0.18 ***
Total difficulties	−0.02	0.02	−0.11 *
Subjective well-being	0.14 **	−0.06	0.46 ***
Prosocial behavior	−0.14 **	−0.04	−0.33 ***
Bipolar experience	−0.05	0.04	0.16 ***
Suicide risk	0.10 *	−0.10 *	

* *p* < 0.05, ** *p* < 0.01, *** *p* < 0.001. a Spearman correlation, b Pearson correlation.

**Table 3 ijerph-18-03024-t003:** Mediation model of total difficulties and subscales of difficulties and well-being on the relationship between bipolar experiences and suicide risk (*N* = 467).

Paths between Variables	Path Symbols	Coefficient *SE*		*p*	Bootstrap 90% CI	
Total difficulties						
Bipolar experiences → Total difficulties	a1	0.88	0.17	<0.001	0.600	1.15
Bipolar experiences → Subjective well-being	a2	−0.62	0.29	<0.05	−1.09	−0.14
Total difficulties → Suicide risk	b1	0.10	0.01	<0.001	0.08	0.12
Subjective well-being → Suicide risk	b2	−0.03	0.01	<0.001	−0.05	0.02
Bipolar experiences → Suicide risk	c	0.06	0.05	0.20	−0.02	0.14
Gender → Suicide risk	c1	0.01	0.12	0.92	−0.19	0.21
Gender X Total difficulties → Suicide risk	c2	0.01	0.03	0.63	−0.03	0.06
Gender X Subjective well-being → Suicide risk	c3	−0.01	0.02	0.78	−0.03	0.02
Age (covariate)		−0.10	0.05	<0.05	−0.18	−0.03
		R^2^ = 0.21		<0.001		
Emotional symptoms						
Bipolar experiences → Emotional symptoms	a1	0.17	0.07	<0.05	0.05	0.29–0.15
Bipolar experiences → Subjective well-being	a2	−0.62	0.28	<0.05	−1.07	0.27
Emotional symptoms → Suicide risk	b1	0.22	0.03	<0.001	0.18	−0.02
Subjective well-being → Suicide risk	b2	−0.04	0.01	<0.05	−0.05	0.19
Bipolar experiences → Suicide risk	c	0.11	0.05	0.08	0.02	−0.01
Gender → Suicide risk	c1	−0.21	0.12	0.23	−0.41	0.03
Gender X Emotional symptoms→ Suicide risk	c2	−0.07	0.06	0.47	−0.16	0.02
Gender X Subjective well-being → Suicide risk	c3	−0.01	0.02	<0.01	−0.04	−0.06
Age (covariate)		−0.13	0.04	<0.001	−0.20	
		R^2^ = 0.24				
Behavioral problems						
Bipolar experiences → Behavioral problems	a1	0.17	0.07	<0.05	0.05	0.29
Bipolar experiences → Subjective well-being	a2	−0.62	0.28	<0.05	−1.07	−0.15
Behavioral Problems → Suicide risk	b1	0.22	0.03	<0.001	0.18	0.27
Subjective well-being → Suicide risk	b2	−0.04	0.01	<0.001	−0.05	−0.02
Bipolar experiences → Suicide risk	c	0.11	0.05	<0.05	0.02	0.19
Gender → Suicide risk	c1	−0.21	0.12	0.08	−0.41	−0.01
Gender X Behavioral Problems → Suicide risk	c2	−0.07	0.06	0.23	−0.16	0.03
Gender X Subjective well-being → Suicide risk	c3	−0.01	0.02	0.47	−0.04	0.02
Age (covariate)		−0.13	0.04	<0.01	−0.20	−0.06
		R^2^ = 0.24		<0.05		
Peer problems						
Bipolar experiences → Peer problems	a1	0.17	0.06	<0.01	0.07	0.27
Bipolar experiences → Subjective well-being	a2	−0.62	0.29	<0.05	−1.09	−0.14
Peer problems → Suicide risk	b1	0.21	0.05	<0.001	0.13	0.29
Subjective well-being → Suicide risk	b2	−0.04	0.01	<0.001	−0.05	−0.03
Bipolar experiences → Suicide risk	c	0.11	0.05	<0.05	0.03	0.19
Gender → Suicide risk	c1	0.15	0.13	0.24	−0.06	0.35
Gender X Peer problems→ Suicide risk	c2	0.07	0.10	0.47	−0.10	0.24
Gender X Subjective well-being → Suicide risk	c3	−0.01	0.02	0.80	−0.03	0.02
Age (covariate)		−0.14	0.05	<0.01	−0.21	−0.06
		R2 = 0.16		<0.001		
Hyperactivity						
Bipolar experiences → Hyperactivity	a1	0.34	0.06	<0.001	0.24	0.44
Bipolar experiences → Subjective well-being	a2	−0.62	0.28	<0.05	−1.08	−0.15
Hyperactivity → Suicide risk	b1	0.08	0.03	<0.01	0.03	0.12
Subjective well-being → Suicide risk	b2	−0.05	0.01	<0.001	−0.07	−0.04
Bipolar experiences → Suicide risk	c	0.12	0.05	<0.5	0.03	0.20
Gender → Suicide risk	c1	0.10	0.13	0.43	−0.11	0.31
Gender X Hyperactivity → Suicide risk	c2	−0.02	0.06	0.80	−0.12	0.09
Gender X Subjective well-being → Suicide risk	c3	−0.02	0.02	0.31	−0.05	0.01
Age (covariate)		−0.13	0.04	<0.01	−0.20	−0.05
		R^2^ = 0.16		<0.001		
Prosocial behavior						
Bipolar experiences → Prosocial behavior	a1	−0.09	0.05	0.10	−0.17	
Bipolar experiences → Subjective well-being	a2	−0.62	0.29	<0.05	−1.09	−0.13
Prosocial behavior → Suicide risk	b1	−0.04	0.05	0.43	−0.11	0.04
Subjective well-being → Suicide risk	b2	−0.05	0.01	<0.001	−0.07	−0.04
Bipolar experiences → Suicide risk	c	0.14	0.05	<0.01	0.05	0.22
Gender → Suicide risk	c1	0.12	0.13	0.35	−0.09	0.33
Gender X Prosocial behavior → Suicide risk	c2	0.02	0.10	0.88	−0.15	0.18
Gender X Subjective well-being → Suicide risk	c3	−0.02	0.02	0.26	−0.05	0.01
Age (covariate)		−0.13	0.05	<0.01	−0.21	−0.06
		R^2^ = 0.15		<0.001		

## Data Availability

Supplementary Materials associated with this article can be found, in the online version, at doi:10.1016/j.psychres.2018.10.043.

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
