# Peer review of "Adolescents’ Bipolar Experiences and Suicide Risk: Well-being and Mental Health Difficulties as Mediators"

_ijerph, 2021, doi:10.3390/ijerph18063024_

Round 1

Reviewer 1 Report

This article is about the identification of the risk and protective factors involved in suicide risk among adolescence, focusing in particular on bipolar experiences. The results indicated that the effect of bipolar experiences on suicide risk is mediated by emotional problems, problems with peers, behavior problems and difficulties associated with hyperactivity. 

The article is original and is conducted with a rigorous statistical methodology. On the other hand, I found some problems in the drafting of the introduction and some potentially critical methodologic issues. In particular about the definitions adopted by the authors of bipolar experiences, that possibly affected the results. 

Here are some comments for the authors.

Introduction

The authors state to predict suicide risk but being this a cross-sectional study, this term is incorrect. At the most, they could say to have found associations between the variables. 

To me, in general the introduction does not flow linearly. I suggest the authors to reorganize this part of the paper to better present the different topics they want to introduce (e.g. adolescence, suicide risk, bipolar disorder) in a more logical way. This would help the reader to have an accurate idea of what you are going to present further.

“Suicide attempts are associated with the female gender, younger age at onset”: what this “onset” refers to? This is not clear.

“In Spain, approximately 4% of a sample of children and adolescents evaluated through the Spanish National Health Survey presented some type of emotional or behavioral problems”: some data about other countries would be useful here.

“Also, adolescents spend more time with their peers than with their family”: this seems to me an arbitrary statement.

“In our study, bipolar experiences or symptoms are defined as the occurrence of seven or more items of the 13 proposed in a self-report measure of manic episodes”: why did the authors decide to consider only manic symptoms in their definition of bipolar experiences? Why they did not consider the depressive side of the disorder? Is there some reference to sustain this choice? This is a major concern that authors need to explain.

Methods

“The final sample was 1,506 students from 14 to 19 years of age (M = 16.15, SD = 1.36). In the sample, 55.7% were females. Of the total participants, 31% (N = 467) had bipolar experiences or symptoms. In this group with bipolar experiences, the mean age was 16.22 (SD = 1.36) and 54.2% were females. Of the total, 50.7% answered affirmatively at least one item of the suicide risk scale. Specifically, 46.9% reported suicidal ideation, 26.6% had planned suicide and 6% reported suicide attempts. With regards to nationality, the distribution was as follows: 89.9% Spanish, 3.7% Latin American (Bolivia, Argentina, Colombia, and Ecuador), 0.7% Portuguese, 2.4% Romanian, 1% Moroccan, 0.7% Pakistani, and 2% other nationalities.”: this part better belongs to the results section. Moreover 46.9% is a very high result. How many positive items were needed for speaking of suicidal ideation? Authors should specify this.

“In this study, to classify the adolescents with bipolar experiences or symptoms, only criterion 1 was considered”: why the authors decided not to consider the other criteria? Maybe it would be better to consider at least also criterion 3, to not overestimate manic symptoms, even if I suppose this is not changeable now.

Results

Table 1 is not cited in the text. Moreover there are no descriptions for the columns. To me the table is completely unclear.

Discussion

“The gender differences reflect the way in which maladjustment is expressed. In females it takes the form of internalizing problems, which could be associated with greater vulnerability to depression, and in males it takes the form of externalizing problems, which is associated with behavioral disorders”: this statement seems to me a bit arbitrary. Authors should try to find more references to support their idea. 

“Also, a narrower definition of bipolar disorder would reflect lower prevalence”: it seems to me that authors considered only the manic side of the bipolar disorder. Why they insist to sustain to have considered bipolar disorder?

Reviewer 2 Report

Overall, this is an interesting study and well-written manuscript that has the potential to shed more light on identifying the risk and protective factors involved in suicide risk in adolescents. The “Introduction” section provided a reader with the necessary background and details for interpreting the results. Generally, statistical methods were adequately used. The “Discussion” section was suitable and wide. Moreover, large and new literature was referenced. However, some points require further attention:
- I propose to add to the keywords "adolescents".
- Which data about the type of emotional or behavioral problems are from other countries?
I propose to read:
Murray, C. J., Abbafati, C., Abbas, K. M., Abbasi, M., Abbasi-Kangevari, M., Abd-Allah, F., ... & Nagaraja, S. B. (2020). Five insights from the global burden of disease study 2019. The Lancet, 396(10258), 1135-1159.
Murray, C. J., Aravkin, A. Y., Zheng, P., Abbafati, C., Abbas, K. M., Abbasi-Kangevari, M., ... & Borzouei, S. (2020). Global burden of 87 risk factors in 204 countries and territories, 1990–2019: a systematic analysis for the Global Burden of Disease Study 2019. The Lancet, 396(10258), 1223-1249.
- In the "Participants" section should be information, that the authors were analyzing data of two groups of adolescents without and with bipolar experiences.
- In the "Statistical data" section should be informed about the test of normality of the variables (e.g., Kolmogorov–Smirnov test, skewness, and kurtosis).
- Moreover, there is a lack of information about MANOVA (what was intra-objective and inter-objective factors).
- In Table 1, there is a lack of the subtitles of variables (the first line).

Reviewer 3 Report

This manuscript entitled "Adolescents’ bipolar experiences and suicide risk: well-being and mental health difficulties as mediators" aimed to identify the risk and protective factors involved in suicide risk.

The manuscript is very interesting with a well-conducted study design with a large and representative sample size. However, some issues should be addressed by the authors:

ABSTRACT

  • include the percentage after '467'
  • include some information about the instruments/questionnaires...

INTRODUCTION

  • Please, avoid to write a paragraph with just one sentence. Take this suggestion for all manuscript. Each paragraph should has a introduction, develoption and conclusion.

METHODS

  • Include clearly the study design in the begining of the method's section
  • It is not well described if all instruments are validated for Spanish language and for Spanish population. Please, follow for all instruments as you did for the PSS instrument: "The psychometric properties of the Spanish PSS".

RESULTS

  • For me the table 3 is confused and could be improved for better understanding.

DISCUSSION

  • Please, refer in the discution section the how essential is the developing of longitudinal studies to improve the understanding of this topic.
  • Please, reduce the conclusion section to be more objective and answer to the aim.

REFERENCES

  • Several recent articles from IJERPH could be cited to improve the text.
  • Below I suggest some articles about mental health in adolescents which may improve the introduction and discussion sections:

Escobar, D.F.S.S.; Noll, P.R.S.; Jesus, T.F.; Noll, M. Assessing the Mental Health of Brazilian Students Involved in Risky Behaviors. Int. J. Environ. Res. Public Health 202017, 3647.

Seven, Ü.S.; Stoll, M.; Dubbert, D.; Kohls, C.; Werner, P.; Kalbe, E. Perception, Attitudes, and Experiences Regarding Mental Health Problems and Web Based Mental Health Information Amongst Young People with and without Migration Background in Germany. A Qualitative Study. Int. J. Environ. Res. Public Health 202118, 81.

Tahara, M.; Mashizume, Y.; Takahashi, K. Coping Mechanisms: Exploring Strategies Utilized by Japanese Healthcare Workers to Reduce Stress and Improve Mental Health during the COVID-19 Pandemic. Int. J. Environ. Res. Public Health 202118, 131.

Escobar, D.F.S.S.; Jesus, T.F.; Noll, P.R.S.; Noll, M. Family and School Context: Effects on the Mental Health of Brazilian Students. Int. J. Environ. Res. Public Health 202017, 6042.

Round 2

Reviewer 3 Report

All my comments were correctly addressed by the authors. Congrats.